# Understanding the Effect of Different Glucose Concentrations in the Oligotrophic Bacterium *Bacillus subtilis* BS-G1 through Transcriptomics Analysis

**DOI:** 10.3390/microorganisms11102401

**Published:** 2023-09-26

**Authors:** Liping Chen, Chenglong Wang, Jianyu Su

**Affiliations:** Key Laboratory of Ministry of Education for Protection and Utilization of Special Biological Resources, School of Life Sciences, Ningxia University, Yinchuan 750021, China; nxu_clp@163.com (L.C.); 13037990903@163.com (C.W.)

**Keywords:** oligotrophic bacteria, low glucose tolerance, sRNA, transcriptome

## Abstract

Glucose is an important carbon source for microbial growth, and its content in infertile soils is essential for the growth of bacteria. Since the mechanism of oligotrophic bacterium adaptation in barren soils is unclear, this research employed RNA-seq technology to examine the impact of glucose concentration on the oligotrophic bacterium *B. subtilis* BS-G1 in soil affected by desertification. A global transcriptome analysis (RNA-Seq) revealed that the significantly differentially expressed genes (DEGs) histidine metabolism, glutamate synthesis, the HIF-1 signaling pathway, sporulation, and the TCA cycle pathway of *B. subtilis* BS-G1 were significantly enriched with a 0.015 g/L glucose concentration (L group), compared to a 10 g/L glucose concentration (H group). The DEGs amino acid system, two-component system, metal ion transport, and nitrogen metabolism system of *B. subtilis* BS-G1 were significantly enriched in the 5 g/L glucose concentration (M group), compared with the H group. In addition, the present study identified the regulation pattern and key genes under a low-glucose environment (7 mRNAs and 16 sRNAs). This study primarily investigates the variances in the regulatory pathways of the oligotrophic *B. subtilis* BS-G1, which holds substantial importance in comprehending the mechanism underlying the limited sugar tolerance of oligotrophic bacteria.

## 1. Introduction

The limitation of nutrients in ecosystems is a significant constraint on the vitality of microbial life [1]. Numerous microorganisms that thrive in nutrient-poor environments, including freshwater lakes, deserts, plateaus, and oceans, have been identified [2,3,4,5]. Among these microorganisms, nutrient-poor bacteria can be classified as specialized or parthenogenetic. Obligately oligotrophic bacteria can only proliferate in media containing carbon concentrations ranging from 1 to 15 mg/L and are not well-suited to nutrient-rich conditions [6,7]. Conversely, facultatively oligotrophic bacteria are capable of growth and adaptation in both high and low concentrations of organic carbon [8]. Due to their predominant distribution in highly nutrient-depleted environmental settings, this particular assemblage of bacteria exhibits sluggish growth rates, protracted growth cycles, a diminutive size, and the ability to traverse 0.45 μm filter membranes. Consequently, researchers have designated them as filterable bacteria and ultramicrobacteria [9,10]. Given their exacting requirements for survival conditions, oligotrophic bacteria are difficult to culture in the laboratory [11]. For example, the discovery of *Planctomycetes* can be attributed to the Hungarian biologist Nador Gimesi who, in 1924, initially found it at the lake Langymanyos. However, due to the unavailability of a pure culture strain, researchers worldwide were limited to studying it solely through its morphological description. It was not until 1973 that Staley successfully isolated the strain, marking a significant milestone in the understanding of *Planctomycetes* [12]. In 1976, Bauld and Staley provided a formal description of this strain as the inaugural species within the phylum *Planctomycetes*, relying on phenotypic and genetic characterization. Subsequently, from 1987 to 2006, Carl Woese and Strous et al. further substantiated the relationship between *Planctomycetes* and *Chlamydomonas* through phenotypic observations and comparisons of protein sequences [13]. The precise identification of the strain took a total of 76 years (1924–2020), with 49 of those years (1924–1973) attributed to the inability to acquire a pure culture of the strain, which hindered its comprehensive study by microbiologists [14].

Bacteria are required to swiftly acclimate to ongoing fluctuations in their surroundings, encompassing scarcities in nutrients, variations in oxygen availability, and exposure to abiotic stresses (e.g., alterations in temperature) [7,15,16]. This adaptability is especially crucial for bacteria that exist independently. Glucose is the optimum carbon source for bacteria, as it supplies both the energy and material resources necessary for strain growth. Therefore, a diminished concentration of glucose hurts strain growth [9,17,18]. Previous studies have demonstrated that propionic acid, a metabolite known for its growth-inhibitory properties, accumulates significantly in a culture medium under controlled glucose concentrations of 1 g/L and 0.2 g/L, consequently impacting the growth conditions of *B. subtilis* [10]. Additionally, it has been proposed that the presence of β-glucosidase (UnBgl1A) enables *B. subtilis* to withstand a glucose concentration of 0.9 M [9,19]. However, previous research investigating the adaptation of *B. subtilis* to low-sugar conditions employed glucose concentration as a manipulated variable to examine its impact on strain growth rate, the production of fermentation products (such as microbial fibrillases), the feedback inhibition of β-glucosidase, and glucose transport [9,10,19,20]. The phosphoenolpyruvate-dependent phosphotransferase system (PTS) is of the utmost importance in the bacterial glucose transportation process, facilitating the movement of specific sugars across the bacterial inner membrane through sugar phosphorylation [21,22,23,24,25]. It has been observed that bacteria commonly possess a PTS for glucose uptake, and it has been discovered that a non-glucose PTS can transport glucose as well (e.g., the cellobiose PTS can facilitate intracellular glucose transport) [22]. Another study revealed that the PTS is capable of detecting variations in nutrient levels within the surrounding environment, thereby facilitating appropriate cellular responses to these fluctuations [26].

Bacterial small regulatory RNAs (sRNAs), which typically range from 50 to 500 nucleotides in length, serve as significant post-transcriptional regulators of gene expression in response to environmental stimuli stressors [27,28,29,30,31,32]. sRNAs, which are transcribed from intergenic regions and do not undergo translation, exhibit a high degree of conservation among homologs [33]. These sRNAs exert their influence on mRNA stability and translation by forming base pairs with the 5′-untranslated region (5′-UTR) of the target mRNA or with bases in the ribosome-binding site (RBS) [28,34,35,36]. sRNAs can positively or negatively regulate target genes. For example, sRNAs can negatively regulate their targets by inhibiting translation or stimulating degradation via ribonuclease RNase E [37,38,39,40]. The interaction of sRNAs with target mRNAs generally requires the RNA chaperone Hfq, which binds to sRNAs, promotes sRNA–mRNA base pairing, and directly binds and regulates the translation of certain mRNAs [39,41,42,43]. Furthermore, one sRNA usually regulates multiple mRNAs, and one mRNA can be handled by various sRNAs, thus forming a regulatory network to respond to changing environments [44]. sRNAs have been identified in many bacteria, including *Escherichia coli, Staphylococcus aureus*, and *Vibrio cholerae*, where they play key roles in these bacteria’s adaptation to environmental stress and virulence [41,42,45]. sRNAs also affect the pathogenesis of Gram-positive *Enterococcus faecalis* by regulating its growth and survival under various environmental stresses, including iron stress [43].

Many researchers have studied the responses of oligotrophic bacteria to low nutrition. *Prochlorococcus*, a *cyanobacterium*, exhibits a high abundance in oligotrophic marine environments [46]. Studies have shown that nitrogen limitation reduces the photosynthetic rate of *Prochlorococcus*, while it increases the concentration of metabolic intermediates [47]. Researchers have found that *Erwinia amylovora* responds to oligotrophic environments by reducing its cell volume and regulating the expression of genes related to hunger, oxidative stress, motility, pathogenicity, and virulence [48]. Furthermore, *Prochlorococcus* can decrease protein abundance and suppress gene translation under conditions of nitrogen limitation, consequently diminishing the cell’s nitrogen requirements [49]. *B. subtilis* is characterized by a fast growth rate and low nutritional requirements. Under unfavorable growing conditions and nutrient deficiency, the organism typically enters a period of spore dormancy, during which it develops spores that exhibit a robust resilience to adverse circumstances. *B. subtilis* exhibits multiple coping strategies in response to abiotic stresses such as ethanol, alkali, low-nutrient, and nitrogen stresses. *B. subtilis* mitigates the deleterious effects of salt stress on the strain through the modulation of antioxidant defense and the glyoxalase system, alongside the maintenance of ion homeostasis and osmotic regulation [50]. The synthesis of tryptophan in *B. subtilis* is enhanced under conditions of ethanol stress [51]. In the presence of alkaline stress, the biomass of *Bacillus natto* exhibited an increase, while the rate of spore formation demonstrated a decrease. Additionally, the synthesis and accumulation of menaquinone-7 (MK-7) were facilitated [52]. The oligotrophic *B. subtilis* cells showed an increased tolerance to chloramphenicol, ampicillin, and oxidative stress [53]. However, researchers have discovered that, when subjected to severe hunger conditions, *B. subtilis* cells exhibit the formation of spherical non-dormant structures, which exhibit prolonged viability for several months [53]. ABC transport-binding proteins are critical for the transport and metabolism of nutrients in oligotrophic bacteria in the ocean [54,55]. In addition, eutrophic bacteria transport external nutrients through the ABC transport system, while oligotrophic bacteria prefer to use the PTS to absorb external nutrients [55,56,57,58]. Oligotrophic bacteria grow slowly in media and pick low-molecular-weight (LMW) organic molecules for growth compared to high concentrations of organic matter (HMW biopolymers) [59,60,61]. Many researchers have explored the hunger stress response of oligotrophic bacteria in water environments. However, there are few reports on the low-sugar tolerance strategies of the oligotrophic bacteria of poor soil, and the mechanism of their adaptation to low sugar is still unclear. Due to the impact of climate change, land use, and human activities, the ecosystem may become unstable, necessitating the implementation of a strategy to detect microbial responses to environmental changes. This approach can serve as an early warning system, enabling the timely detection of environmental changes and facilitating the implementation of appropriate remedial measures. Research has demonstrated that the assessment of *cyanobacteria* density in aquatic environments can effectively predict alterations in water quality, thereby assisting managers in mitigating water quality issues [62]. In addition, researchers have used qRT-PCR to quantify *Microcystis* spp. toxin genes in lakes to determine the dynamics of *cyanobacterial* blooms [63,64]. Ningxia is one of the provinces with the most severe desertification in China, with a desertification area of 2.898 million hm^2^, accounting for 55.8% of the total land area of Ningxia. Land degradation has seriously affected the local ecological environment and people’s lives [19,65]. Hence, the accurate anticipation of alterations in soil nutrient levels holds significant relevance in the context of managing sandy soils. This study employed RNA-Seq to elucidate the adaptation patterns of *B. subtilis* BS-G1 under varying glucose concentrations. In addition, this study aims to identify the crucial mRNAs and sRNAs involved in low-glucose adaptation and establish a theoretical foundation for the development of biogenetic early warning indicators for soil impoverishment.

## 2. Materials and Methods

### 2.1. Strain

The oligotrophic bacterium of *B. subtilis* BS-G1 was isolated from microbial crusts that developed early in the desertification area of Ningxia. *B. subtilis* W1, *B. subtilis* MX31, *B. subtilis* ACCC11025, *B. subtilis* 168, and *B. subtilis* L1 were stored in our laboratory (Ningxia University, Yinchuan, China).

### 2.2. Growth Curve and Determination of Residual Sugar Content

One loop of each strain was taken from an agar plate, inoculated in 5 g/L oligotrophic culture medium, and incubated at 37 °C, 150 r/min. After approximately 8 h, the bacterial concentration reached approximately 1.2 (OD_600_). This was reserved as seed solution. The seed solution was inoculated in an oligotrophic medium with 3 glucose concentrations (0.015 g/L, 5 g/L, and 10 g/L) and cultured at 37 °C and 150 r/min for the growth curve. The bacterial residual sugar content in the medium was measured every 2 h using a biosensor analyzer, SBA-40D.

### 2.3. Construction of Transcriptome and Processing of Biological Information

The activated *B. subtilis* BS-G1 seed solution was picked and inoculated with 2% inoculum in the oligotrophic culture medium, cultured at 37 °C, 150 r/min for 8 h. The bacterial mixture was then centrifuged at 10,000× *g* for 15 min to obtain the total bacterial pellets. The pellets were gently washed three times with PBS (10 mM, pH 7.4) and all experiments were conducted with three biological replicates. The Trizol reagent method was employed to extract total RNA from the collected cells, while DNase I (Takara, Beijing, China) was used to remove genomic DNA.

Fragmented mRNA was used as a template to synthesize the first strand of cDNA, and RNase H was used to degrade the RNA strand for the preparation of the second strand of cDNA. The purified cDNA was repaired and connected to the sequencing adapter, and the library was completed after purification. RNA-seq paired-end sequencing was performed using Illumina HiSeq X Ten (2 × 150 bp).

### 2.4. Reading Mapping and Analysis

Read quality was first examined using FastQC. The original read data from RNA-Seq libraries were then mapped to the *B. subtilis* BS-G1 genome, using Bowtie2 with standard parameterization. The calculated uniquely mapped read counts were fed into DESeq2 (version 1.40.2) for the quantitation of significant gene expression with standard parameterization. DEGs identified by DESeq2 were filtered using a moderate absolute |log_2_(Fold Change)| > 1 and *p*-value < 0.05. The read counts were transformed into TPM to evaluate the gene expression levels. The Kyoto Encyclopedia of Genes and Genomes (KEGG) database was used to annotate the lists of significantly expressed genes with pathway details. The Gene Ontology (GO) database predicted the functions of the genes in *B. subtilis* BS-G1.

### 2.5. Bioinformatics Analysis

The bioinformatics analysis was conducted utilizing data generated by the Illumina platform. The analyses were conducted utilizing the cloud platform (cloud.majorbio.com, accessed on 17 August 2020) provided by Shanghai Majorbio Bio-pharm Technology Co., Ltd. (Shanghai, China).

### 2.6. Quantitative Real-Time PCR (qRT-PCR)

qRT-PCR determined the gene expression level. The extracted RNA was used for reverse transcription to obtain cDNA according to the manufacturer’s protocol (Prime Script RT reagent Kit, Takara, Beijing, China). TB Green Premix Ex Taq II (Takara) was utilized to carry out the qRT-PCR reaction. The 2^−ΔΔCT^ method was used to assess the fold change in the target genes compared to the internal reference gene. Each qRT-PCR was performed at least three times. Appendix A includes the primers used for qRT-PCR.

### 2.7. Statistical Analysis

A one-way ANOVA was used for comparisons among multiple groups. *p*-value < 0.05 (*), *p*-value < 0.01 (**), *p*-value < 0.001 (***), and *p*-value < 0.0001 (****) were considered statistically significant, and *p*-value > 0.05 was non-significant (ns). GraphPad Prism was used for statistical analysis.

## 3. Results

### 3.1. Growth Curves and Glucose Utilization

Through the assessment of the growth curve of *B. subtilis* BS-G1 and five control strains under a glucose concentration of 0.015 g/L, it was observed that solely *B. subtilis* BS-G1 exhibited normal growth patterns. *B. subtilis* BS-G1 exhibits normal growth within a glucose concentration range of 0.015–10 g/L, and an elevation in glucose concentration within the culture medium proves advantageous for the strain’s growth (Figure 1). The findings of this study suggest that *B. subtilis* BS-G1 exhibits the characteristics of a facultative oligotrophic bacterium.

By determining the sugar uptake curves of *B. subtilis* BS-G1 and five control strains at a 0.015 g/L glucose concentration, it was found that the fastest rate of sugar uptake was found in *B. subtilis* BS-G1 (Figure 2a). The rate of sugar uptake of *B. subtilis* BS-G1 in the range of 0.015–10 g/L glucose sugar concentration was found to increase with the increase in the glucose concentration, and the strain sugar uptake curve reached a stabilization period at 16 h of incubation (Figure 2b).

### 3.2. Transcriptome Analysis of B. subtilis under Different Glucose Concentrations

In this study, we constructed nine cDNA libraries, including bacteria from the *B. subtilis* BS-G1 strain with three glucose concentrations (L group, M group, H group). Samples were used for processing and controls. After removing sequencing adapters and low-quality data, 592,812,250 filtered reads were obtained with a Q30 base percentage that was higher than 96.11% for each sample. Then, the filtered reads from each sample were aligned to the reference genome, and the filtered reads for each sample were compared with the reference genome. The mapping efficiency was between 98.91% and 99.32%. Higher mapping percentages indicated a good sequencing quality of the samples (Appendix A). A transcriptome data analysis showed that 4020 genes (mRNA) were expressed in *B. subtilis* BS-G1 under different glucose concentrations. The above genes were compared with GO, COG, and KEGG databases. Among them, the COG annotation rate was 77.63%, and the GO annotation rate was 73.25%. The KEGG annotation rate was 54.60% from the perspective of the KEGG pathway (Appendix A). We analyzed the differences in the gene expression of *B. subtilis* BS-G1 treated with different glucose concentrations for 12 h. The results show 1821 differentially expressed DEGs (mRNAs) in the L group vs. the H group (LvsH), of which 1099 DEGs were significantly upregulated and 722 DEGs were significantly downregulated. At the same time, 1571 DEGs were detected in the M group vs. the H group (MvsH), of which 845 DEGs were significantly upregulated and 726 DEGs were considerably downregulated (Appendix A). The DEGs fold change distributions of LvsH and MvsH showed that the genes with the most significant fold change in DEGs were distributed between 1 ≤ |Log_2_FC| < 2 (Appendix A).

### 3.3. Transcriptome Differential Gene Annotation

The DEGs were annotated and classified according to the COG database. As shown in Figure 3, among the six functions, the number of upregulated DEGs is higher than the number of downregulated DEGs. These six functions are “cell cycle control, cell division, chromosome allocation,” “intracellular trafficking, secretion, and vesicular trafficking,” “transcription,” “amino acid transport and metabolism,” “nucleotide transport and metabolism,” and “lipid transport and metabolism” (Figure 3). The annotation results of the DEGs in KEGG pathways showed that the four KEGG functions with the most DEGs in the LvsH and MvsH were “amino acid transport and metabolism,” “carbohydrate transport and metabolism,” “membrane transport,” and “lipid metabolism” (Figure 4). The KEGG annotation results are consistent with previous analyses based on COG categories. The results also show that different sugar levels mainly affected the “amino acid metabolism,” “carbohydrate metabolism,” “membrane transport,” “cofactor and vitamin metabolism,” and “energy metabolism process” of *B. subtilis* BS-G1.

### 3.4. GO Functional Enrichment and KEGG Pathway Enrichment

To systematically determine the DEGs’ possible functions and participating pathways, we performed GO and KEGG pathway enrichment analyses on the DEGs. The GO enrichment analysis showed that the GO annotations of the DEGs were divided into three parts: biological process (BP), cellular composition (CC), and molecular function (MF). It was found that the top 20 GO terms that were enriched by the DEGs of LvsH were mainly divided into three categories, namely transporter activity, sporulation, and catabolism of histidine and acidic organic compounds (Appendix A). The top 20 GO terms that were significantly enriched in DEGs in MvsH were mainly enriched in five aspects: histidine synthesis, nucleotide synthesis and metabolism, oxidoreductase activity, sports, and metabolism of nitrogenous inorganic salts (Appendix A). The corresponding numbers of DEGs in the significantly enriched pathways in LvsH and MvsH are shown in Figure 5.

The KEGG enrichment analysis showed that the DEGs of LvsH were mainly concentrated in the porphyrin and chlorophyll metabolism, PTS, histidine metabolism, glycolysis/gluconeogenesis, butyrate metabolism, and HIF-1 signaling pathway (Figure 6a). The DEGs of MvsH were mainly concentrated in KEGG pathways such as the arginine biosynthesis, flagellar assembly, two-component system, and nitrogen metabolism (Figure 6b).

### 3.5. sRNA Analysis of B. subtilis BS-G1 under Different Glucose Concentrations

A total of 595 sRNAs were identified in the *B. subtilis* BS-G1 strain, and these ranged from 1 nt to 500 nt in length, with an average length of 95 nt (Figure 7b). The sRNAs can be divided into two types: one is the sRNAs that act on the intergenic region of the target mRNA, and the other is the antisense sRNAs that work on the SD sequence or part of the coding region of the target mRNA. Among the sRNAs that were identified in this study, there were 471 intergenic sRNAs and 121 antisense sRNAs, of which 51 were annotated (Figure 7a). Comparing the sRNAs with known functions in the Rfam database, 51 sRNAs were annotated, including 28 rRNAs, 2 tRNAs, 20 CisRNAs, and one which was annotated as a bacterial large signal recognition particle RNA (Figure 7c). In terms of the differential expression analysis display, compared with the H group, the L group had the most differentially expressed sRNAs, including 101 upregulated sRNAs and 41 downregulated sRNAs. The M group had 24 upregulated sRNAs and 30 downregulated sRNAs (Figure 8).

### 3.6. Annotation of Significantly Differentially Expressed sRNAs Target Genes

We compared sRNAs’ target genes that were significantly differentially expressed in LvsH and MvsH with the COG database. The results revealed that the key categories were “carbohydrate transport and metabolism,” “amino acid transport and metabolism,” “transcription,” and “cell wall/membrane/envelope biogenesis” (Figure 9). This is consistent with the COG annotation results for DEGs. In addition, the target genes in the LvsH were also involved in “inorganic ion transport and metabolism,” “replication, recombination and repair,” and “energy production and conversion” (Figure 9a). Target genes were also assigned to “signal transduction mechanisms” and “coenzyme transport and metabolism” in the MvsH (Figure 9b).

### 3.7. Enrichment Analysis of sRNA Target Genes

To further evaluate the function of the significantly differentially expressed sRNAs’ target genes, we performed GO and KEGG enrichment analysis on these target genes. The sRNAs’ target genes that were significantly differentially expressed in LvsH were greatly enriched. The top 20 GO terms in the set were divided into six categories: ATPase activity, hydrolase activity, transporter activity, localization, material transport, and nucleoside triphosphatase activity (Appendix A). The sRNAs’ target gene enrichment pathways of MvsH are classified into 10 categories, namely transporter, nucleoside sugar synthesis, plasma membrane, others, nucleotide synthesis and metabolism, nitrogen metabolism, metal ion transport, ATPase activity, anhydride hydrolase, and amino acid synthesis (Appendix A). The KEGG enrichment results show that LvsH significantly differentially expressed the sRNAs’ target genes and enriched two genes, including those related to photosynthesis and the ABC transporter, as shown in the KEGG pathway (Figure 10a). MvsH significantly differentially expressed the sRNAs’ target genes and enriched five KEGG pathways, namely “RNA polymerase, amino sugar and nucleotide sugar metabolism, purine metabolism, ABC transporter, pentose and glucuronate interconversion” (Figure 10b).

### 3.8. Adaptation Patterns and Key Genes in a Low-Glucose Environment

Based on the enrichment results of the *B. subtilis* BS-G1 strain DEGs and the significantly differentially expressed sRNA target genes identified by GO/KEGG, we constructed the adaptation model of the *B. subtilis* BS-G1 strain under different glucose concentrations (Figure 11). As the concentration of glucose in the medium decreased, the activity of the glucose transporter increased, the aerobic oxidation pathway of glucose weakened, the amount of ATP produced decreased, and it gradually transformed into a dormant body. In addition, *B. subtilis* BS-G1 in the H group and L group sensed changes in the external nutrient content through the HIF-1 signaling pathway, while *B. subtilis* BS-G1 in the M group received external environmental signals through a two-component system and quickly transmitted them to the bacteria to make more nutritional changes, leading to positive and effective adaptive responses. Nitrogen-containing inorganic salts (NH^4+^ and NO^3−^) and metal ions (Mg^2+^, K^+^, and Ca^2+^) in the culture medium were utilized by strains in the M group. The production and metabolism of amino acids played a positive regulatory role in the strain’s adaptation to different glucose environments, and the output of histidine and arginine in the M group increased. The L group of histidine is converted into glutamic acid, and glutamic acid enters the TCA cycle through further metabolism to generate α-ketoglutarate. Among the amino acid groups, the imidazole group of histidine can form coordination compounds with metal ions in the medium, which is beneficial to maintaining cell homeostasis and controlling the transportation of nutrients. The highly helical flagella of the strains in the M group increased the motility, driving the strains to more beneficial nutritional small molecules or ions and avoiding substances that were not conducive to their growth. At different glucose concentrations, sRNAs’ targets regulate transporter activity, ion transport, amino acid production, and ATP generation pathways.

### 3.9. Critical Genes for Low-Sugar Adaptation

To explore the adaptation mechanism of *B. subtilis* BS-G1 under a low concentration of glucose, we determined the vital pathway for the strain to adapt to an environment of 0.015 g/L glucose through an GO/KEGG enrichment analysis of the DEGs and sRNA target genes of LvsH, namely the transport pathway. All genes in the transport pathway were screened (screening condition |Log_2_FC| ≥ 5), and, finally, crucial mRNAs (7) and sRNAs (15) that were adapted to the 0.015 g/L glucose environment were obtained. The seven key mRNAs were divided into three categories, amino acid ABC transporters, PTS component proteins, and glutaminase ABC transporters (Table 1 and Table 2).

Taking seven key mRNAs as the central genes and analyzing their interaction with their corresponding sRNAs, the two key mRNAs (gene0435 and gene3876) with the highest degree of network connection were determined. *mtlA* (gene0435) and *celB* (gene3876) are mannitol and mannose PTS transporter subunits, respectively. Among them, three key sRNAs (sRNA0504, sRNA0362, and sRNA0200) were connected to two key mRNAs (Figure 12).

### 3.10. qRT-PCR Validation of DEGs

To further verify the reliability of the transcriptome results, we selected five DEGs (gene3877, gene0435, gene0646, gene1536, gene0436) that are related to carbohydrate transport for qRT-PCR (Appendix A). Although the relative expression values determined by qRT-PCR did not exactly match those obtained from transcriptome sequencing, they were similar to the gene expression trends (up or downregulation) obtained from the transcriptomes.

## 4. Discussion

It has been shown in previous reports that the transport system of bacteria is closely related to stress [66,67]. In this study, we explored possible mechanisms of sugar stress through transcriptome analysis. This study showed that the four pathways and PTS related to transporter activity were significantly enriched in the LvsH by GO functional enrichment and KEGG enrichment (Figure 5 and Figure 6). It was reported that the transporter activity genes of *Lactiplantibacillus plantarum* ZDY2013 were significantly differentially expressed under acid stress [66]. This is consistent with the results of this study. In this study, the DEGs associated with LvsH transporter activity were mainly divided into ABC transporters, MFS transporters, and PTS transporters. Studies have shown that, under acid-stress conditions, genes related to ABC and PTS transporters in *L. plantarum* ZDY2013 changed significantly [66]. The MFS transporter is the most important secondary active transporter on the cell membrane, and can selectively transport monosaccharides, oligosaccharides, nucleotides, and other substrates. Studies have shown that, under pH stress, the expression of the MFS transporter of *Enterococcus faecalis* is upregulated, thereby enhancing the efflux capacity of hydrogen ions and enhancing the strain’s tolerance to acid [67]. Glucose is mainly transported across membranes through the PTS. Studies have shown that, under butanol stress, the transcription of PTS genes transporting mannitol and cellobiose in *L. plantarum* WCFS1 is downregulated [68]. It has been reported that, under oxidative stress, the expression of multiple PTS genes in *Listeria monocytogenes* YjbH is downregulated, and the degree of downregulation is as great as hundreds of times the normal amount [69]. In addition, transport proteins are important osmoregulators that are responsible for the uptake and excretion of vital substances such as inorganic ions, sugars, and amino acids. This plays a critical role in regulating the osmotic pressure of the *B. subtilis* BS-G1 strain in a low-concentration glucose environment.

Metal ions play an essential role in maintaining cellular homeostasis and controlling transport. Among them, K^+^ and Na^+^ play a vital role in cell homeostasis, membrane transport, and the regulation of osmotic pressure [70,71,72]. Ktr system potassium transporter A (gene3042) was significantly upregulated in both LvsH and MvsH, and the degree of upregulation of LvsH was more prominent (Appendix A). There are two forms of Na^+^ transporter, symport and anti-transport. Na^+^ symport may be related to the absorption of amino acids, sugars, organic cations, or anions. The sodium–proton antiporter (gene0984) was significantly upregulated in both LvsH and MvsH. Studies have shown that, when *Dietzia* sp. DQ12-45-1b after Na^+^/H^+^ antiporter mutation and wild-type *Dietzia* sp., DQ12-45-1b were cultured in 0.25–1 M NaCl. Simultaneously, in cells of the wild-type strain, the degree of enrichment was significantly higher than that of the mutant strain [73]. K^+^ and Na^+^ are closely related to the spore germination process, and spore germination will stimulate the release of K^+^ and Na^+^ [74]. In addition, studies have shown that the accumulation of K^+^ also induces the germination of spores [75]. In this study, “endospore-forming prespores” and “intracellular immature spores” were highly enriched among GO functional enrichments in DEGs of the LvsH. Under harsh conditions, *B. subtilis* can be transformed into endospores to survive under the regulation of SigB [76]. When favorable growing conditions return, the spores exit dormancy and germinate. Mg^2+^ is an essential divalent cation for every cell and an important cofactor in DNA replication, transcription, and translation [77,78,79]. Mg^2+^ must pass through the biomembrane through Mg^2+^ transporters. So far, there are four Mg^2+^ transporters: CorA, MgtA/B, MgtA, and NramP [80]. However, in prokaryotes, Mg^2+^ is mainly transported by CorA and MgtA [77,78,79]. In this study, *corA* (gene0850) was significantly upregulated in both LvsH and MvsH, and the upregulation degree of LvsH was higher than that of MvsH (Appendix A). Jayanti Saha et al. analyzed the genome and comparative genome of Pseudomonas aeruginosa and believed that *corA* might be necessary for strains to resist heavy metal toxicity [81]. It is indicated that, in the L group, *B. subtilis* BS-G1 tried to maintain the homeostasis of cells by strengthening the transport of Mg^2+^ in the medium. Iron ions are essential cofactors for cellular processes, playing important roles in respiration, nitrogen fixation, DNA synthesis, the TCA cycle, and oxygen transport [82,83,84]. As a regulator of iron uptake, Fur can bind to Fe^2+^ to control the vehicle of Fe^2+^ [83,85]. In this study, both LvsH and MvsH iron transporters (gene0417, gene0416, and gene3287) were significantly downregulated, but the degree of downregulation of LvsH was lower than that of MvsH (Appendix A). Transcriptome analysis of the metal ion uptake system of *Clostridium beijerinckii* NRRL B-598 by Maryna Vasylkivska’s team found that the expression of genes related to the iron uptake system increased during the sporulation stage [70]. Iron has been reported to help facilitate the germination process of spores [86]. When Samuel Plante et al. studied spore germination and the siderophore transporter Str1, they believed that siderophores might be secreted when the bacteria transformed from a swollen round reproductive body to a vase-shaped dormant body [86]. As a second messenger, Ca^2+^ can participate in many physiological activities of organisms, including maintaining cell membrane homeostasis, regulating the growth and development of organisms, and regulating enzyme activities [86,87]. Studies have shown that Ca^2+^ helps to reduce the accumulation of cadmium in *Phanerochaete chrysosporium*, thereby reducing the toxic effect of cadmium on the strain [87]. This study showed that the calcium/proton exchanger *chaA* (gene0837) was significantly upregulated in LvsH but not significantly differentially expressed in MvsH. Studies by Yingkun Wan et al. showed that, after chaA-knockout *E. coil* and wild-type *E. coil* were treated with gentamicin for six days, the number of cells in the knockout strain was notably reduced compared to the wild-type strain [88].

Chemotaxis in *B. subtilis* and other bacteria is a widely studied adaptive mechanism by which bacteria detect chemical compounds and exhibit movement towards or away from specific compounds [89,90,91]. This mechanism plays an important role in cell growth, biofilm formation, virulence, and infectivity. The bacterial flagellum is a macromolecular complex consisting of approximately 20,000–30,000 protein subunits, including about 30 different proteins [90]. Our results show that there is a significant upregulation of flagellar synthesis genes in the M group. This study demonstrated that *P. extremaustralis* exhibited the upregulation of flagellar genes (*flgB*, *flgN*, *flgM*) in response to an oxygen-stress environment. This upregulation led to an increased motility of the strain, enabling it to evade toxic compounds such as H_2_O_2_ [92].

sRNAs are regulators involved in gene expression in organisms. Under adverse circumstances, organisms will regulate gene transcription levels through sRNAs or produce new proteins to cope with the stress [93]. In this study, LvsH and MvsH identified 156 and 161 differentially expressed sRNAs’ target genes related to carbohydrate metabolism and transport, respectively (Appendix A). The researchers placed *Salmonella enterica* under non-starvation and carbon-source-starvation treatments and found that the differentially expressed sRNAs’ target genes contained multiple genes related to carbohydrate transport and metabolism [94]. In *Caulobacter crescentus* with oligotrophic characteristics, an sRNA that is closely related to carbon starvation was identified, CrfA, which can target and regulate the mRNA of various membrane transporters [95]. In this study, LvsH and MvsH significantly differentially expressed sRNA target genes, and GO and KEGG functional enrichment showed that ABC transporters, membrane functions, and fatty acid biosynthesis were considerably enriched, indicating that, under different carbon-source conditions, sRNAs may be transported. Protein diversification regulates the strain’s carbon source uptake capacity. In addition, we found that *sacA* (gene3825, gene3444) in LvsH and *sacA* (gene3825), *sacB* (gene4085), and *sacC* (gene4086) in MvsH strains were all significantly upregulated (Appendix A). Sac proteins belong to the Bgl-Sac anti-termination protein family, which is essential in responding to carbon-source starvation. Sac anti-termination protein consists of one RNA-binding domain and two regulatory domains (PRD1 and PRD2), these being reversible phosphorylation binding sites which act in response to cognate carbon sources. In the absence of carbon sources, the protein phosphorylates PRD1 through the EII transporter of the PTS, making it inactive [85]. In the presence of a carbon source, the EII transporter dephosphorylates PRD1, while the HPr protein (a non-sugar-specific component of the PTS) phosphorylates PRD2. When PRD1 is not phosphorylated and PRD2 is phosphorylated, the anti-terminator protein will form a homodimer, which can combine with the target gene to form an anti-terminator structure, thereby ensuring the regular operation of the PTS [96].

Histidine is the α-amino β-imidazolyl propionic acid of the eight-gene operon hisGDC[NB]HAF[IE] [97]. The HisF gene encodes a cyclase that forms a heterodimer holoenzyme imidazole glycerol phosphate synthase (IGPS) with HisH (a transglutaminase), and this enzyme connects three different biological pathways, namely nitrogen metabolism, histidine biosynthesis, and the de novo synthesis of purines [98,99]. At the L group, histidine is converted into glutamic acid under the action of iminomethylglutamic acid, and glutamic acid is converted to glutamic acid by glutamic acid dehydrogenase (GDH), or alanine or aspartic acid transaminases (TAs) convert TCA cycle intermediate α-ketoglutarate (α-KG). After α-KG enters the tricarboxylic acid cycle, it is converted into oxaloacetate, which is converted into phosphoenolpyruvate by phosphoenolpyruvate carboxylase and enters the gluconeogenesis pathway to generate glucose, thus providing the medium with a carbon source. *Coxiella burnetiid* IC can use glutamate as its sole carbon source and metabolize glutamate to produce ketoglutarate and enter the TCA cycle and gluconeogenesis pathway [100]. In addition, the L group strengthens purine metabolism, and purine, as the most abundant metabolic substrate in all organisms, provides essential components for synthesizing DNA and RNA. In addition to being the building blocks of DNA and RNA, purines provide crucial energy and cofactors for cell survival and proliferation. Under the environment of the M group, the glutamine produced by the nitrogen metabolism of *B.subtilis* BS-G1 can enter the histidine synthesis pathway through HisF. The imidazole group of histidine forms a coordination compound with metal ions (Ca^2+^, Mg^2+^, and K^+^), which is beneficial for the strain to maintain cell homeostasis, membrane transport, and the regulation of osmotic pressure. Therefore, the synthesis and metabolism of histidine play a crucial regulatory role in adapting *B.subtilis* BS-G1 to different concentrations.

This study identified seven critical mRNAs that were adapted to the L group, among which *mtlA* (gene0435) and *celB* (gene3876, gene3860) are used for the specific transport of mannitol and cellobiose, respectively [101,102]. Studies have shown that, when the mannitol-specific PTS transporter (*mtlA*) was overexpressed, the strain enhanced the uptake rate of PTS sugars, including N-acetylglucosamine, methyl α-glucoside, and 2-deoxyglucose [103]. The researchers propose that the overexpression of PTS transporter genes specifically enhances the uptake rate of PTS substrates, which may be caused by stimulatory protein–protein interactions between MtlA and target PTS transporters, but the specific mechanism is unclear [103]. In the L group, the strain can transport ammonium ions in the culture medium into the cell, through the glutamine ABC transporter (gene2655 and gene2653), across the membrane [104]. Previous research has demonstrated that the utilization of ABC transporter proteins, which are connected to the transportation of heavy metal ions, can serve as biomarkers for the identification of heavy metal pollution in soil through the examination of their expression [58]. Glucose serves as a primary carbon source for microorganisms, and prior empirical findings have demonstrated that the expression of seven crucial mRNAs related to carbohydrate transport is notably elevated under conditions of low glucose levels. Notably, the genes *mtlA* (gene0435) and *celB* (gene3876, gene3860) are prevalent across various strains, including *Vibrio cholerae*, *Escherichia coli*, *Lactococcus lactis*, *Clavibacter michiganensis*, and *Listeria monocytogenes* [105,106,107,108]. The subsequent research direction of our team involves utilizing *mtlA* (gene0435) and *celB* (gene3876 and gene3860) as bio-alert indicators to assess the nutrient status of soil based on their expression.

## 5. Conclusions

Bacterial adaptation to changing environments is often accompanied by transcriptome remodeling. This study presents evidence of distinct mRNAs’ and sRNAs’ expression patterns between the LvsH and MvsH. Notably, both groups exhibit a significant upregulation of transport proteins, suggesting that *B. subtilis* BS-G1 can adapt to varying glucose concentrations by enhancing its substance transport activity. Furthermore, our investigation identifies seven crucial mRNA and three sRNA molecules that are associated with L-group adaptation. These key mRNA molecules are further classified into three categories: amino acid ABC transporters, PTS component proteins, and glutaminase ABC transporters. This finding has shown that *B. subtilis* BS-G1 exhibits a response to low sugar stress through the upregulation of glucose and amino acid transport, as well as the increased production of glutaminase within the bacterial culture medium. This study developed a model to investigate the adaptation of *B.subtilis* BS-G1 to varying sugar concentrations, focusing on the elucidation of the adaptation process in environments with reduced glucose levels. *B. subtilis* BS-G1 promoted a high expression of PTS transporters (*mtlA* and *celB*) through sRNA targeting in the L group, thereby increasing the rate of glucose uptake by *B. subtilis* BS-G1. In addition, after the histidine produced in the strain is transformed into glutamic acid, the final metabolite α-ketoglutarate produced through further metabolism enters the TCA cycle and the gluconeogenesis pathway successively, providing carbon for the growth of the strain.

The objective of this research was to investigate the impact of varying glucose concentrations on *B. subtilis* BS-G1. Transcriptomics analysis was employed to ascertain the alterations in mRNAs and sRNAs induced by different glucose concentrations, thereby establishing a basis for future investigations. However, this study lacked a verification experiment for seven key mRNAs, and further experimental verification will be conducted by inhibiting the expression of critical genes (mRNAs and sRNAs). This study offers a theoretical foundation for a comprehensive investigation into the mechanism of oligotrophic microbial adaptation in low-sugar environments. Additionally, it identifies potential avenues for future research aimed at further understanding the mechanisms of oligotrophic microbial adaptation in nutrient-depleted soil environments.

## Figures and Tables

**Figure 1 microorganisms-11-02401-f001:**
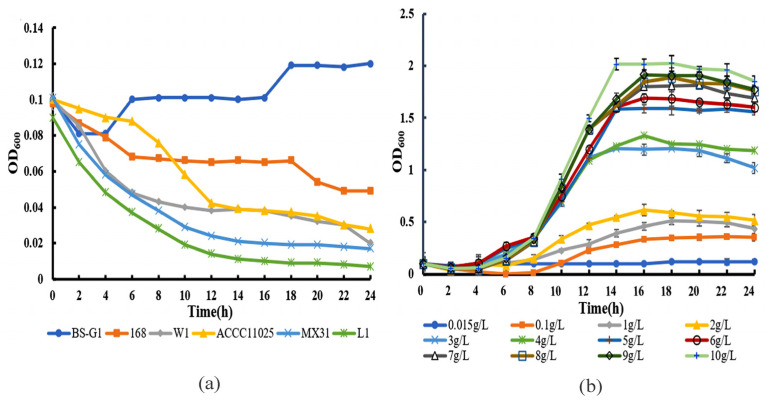
Growth curve. (**a**) Growth curves of *B. subtilis* BS-G1 and five control strains at 0.015 g/L glucose concentration; (**b**) Growth curves of *B. subtilis* BS-G1 at different glucose concentrations.

**Figure 2 microorganisms-11-02401-f002:**
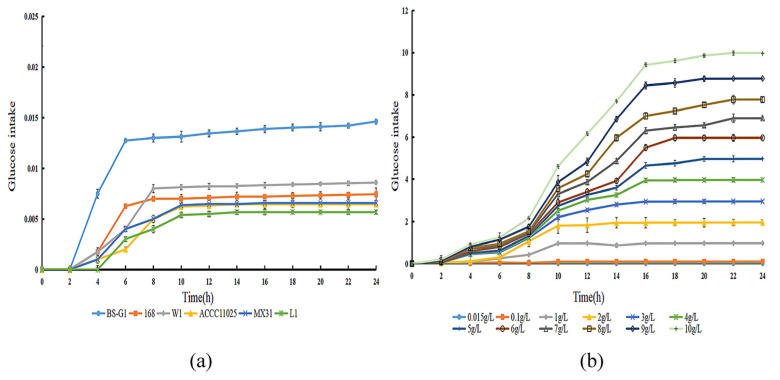
Glucose uptake curve. (**a**) Glucose uptake curves of *B. subtilis* BS-G1 and five control strains at 0.015 g/L glucose concentration uptake curve; (**b**) *B. subtilis* BS-G1 at 0.015–10 g/L glucose concentration glucose uptake curve.

**Figure 3 microorganisms-11-02401-f003:**
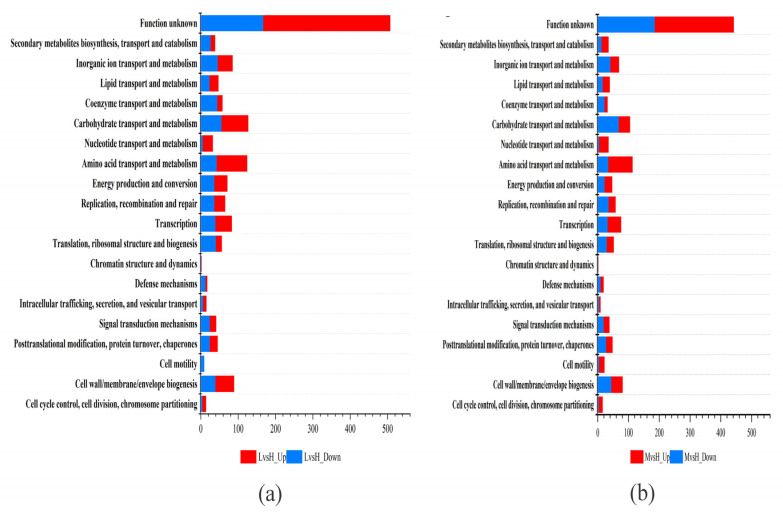
COG functional classification of DEGs. (**a**) COG functional classification results of LvsH; (**b**) COG functional classification results of MvsH.

**Figure 4 microorganisms-11-02401-f004:**
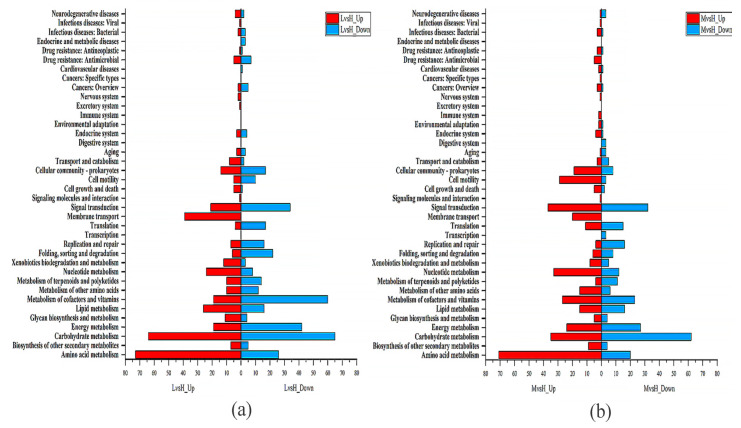
KEGG functional classification of DEGs. (**a**) KEGG functional classification results of LvsH; (**b**) KEGG functional classification results of MvsH.

**Figure 5 microorganisms-11-02401-f005:**
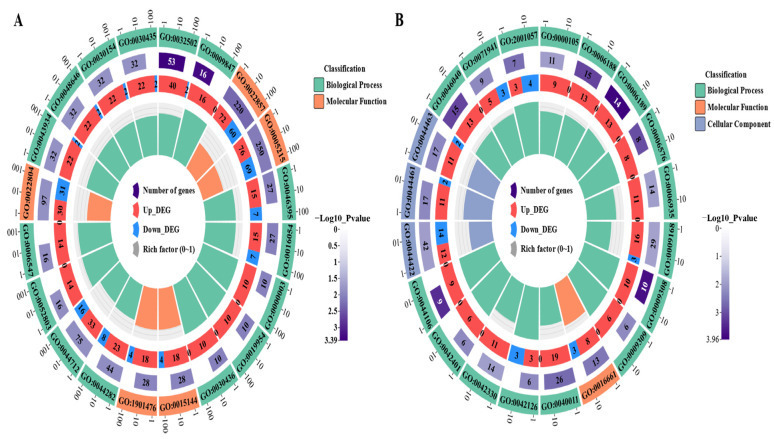
GO enrichment analysis results of DEGs. (**A**) GO enrichment results of LvsH; (**B**) GO enrichment results of MvsH (the diagram of GO enrichment: from outside to inside, the outermost circle is the ID number of GO enrichment, the second circle is the background gene, the third circle is the number of DEGs, and the fourth circle is the enrichment factor. That is, the number of DEGs divided by the number of background genes).

**Figure 6 microorganisms-11-02401-f006:**
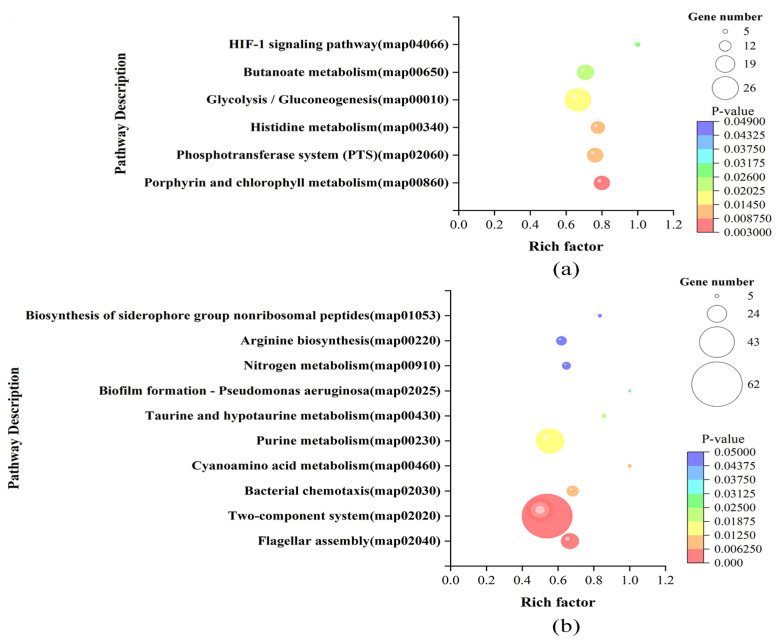
KEGG enrichment analysis results of DEGs. (**a**) KEGG enrichment results of LvsH; (**b**) KEGG enrichment results of MvsH.

**Figure 7 microorganisms-11-02401-f007:**
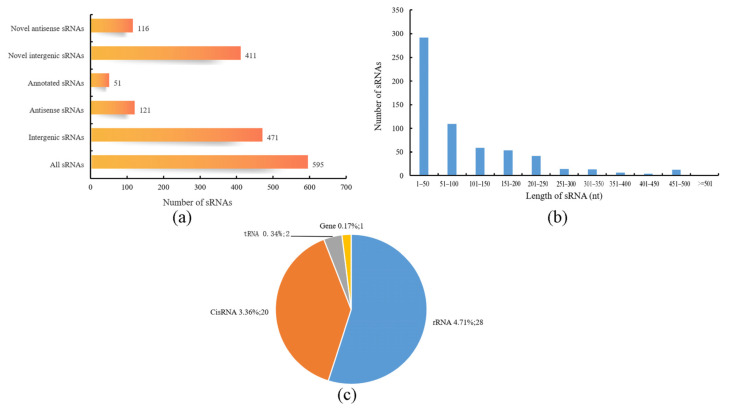
Characteristics of sRNAs. (**a**) Statistics of cis-acting sRNAs and trans-acting sRNAs; (**b**) sRNA length distribution; (**c**) Rfam annotation statistics of sRNAs.

**Figure 8 microorganisms-11-02401-f008:**
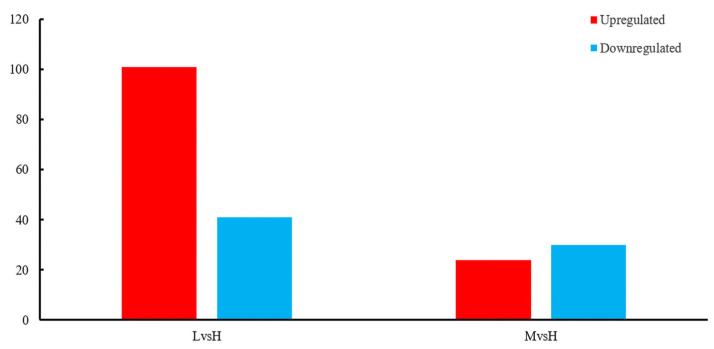
Significantly differentially expressed sRNAs.

**Figure 9 microorganisms-11-02401-f009:**
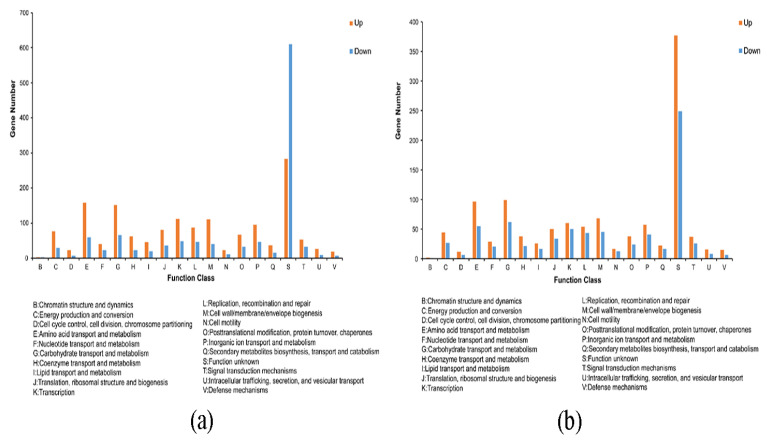
COG annotation classification of target genes with significant differential expression of sRNAs. (**a**) COG annotation classification results of LvsH; (**b**) COG annotation classification results of MvsH.

**Figure 10 microorganisms-11-02401-f010:**
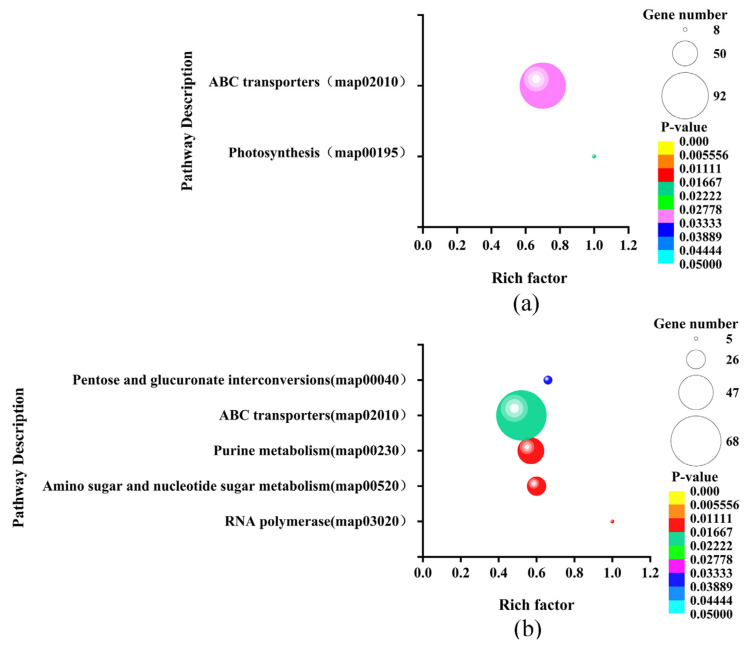
KEGG enrichment of significantly differentially expressed sRNAs target genes. (**a**) KEGG enrichment of target genes of LvsH differentially expressed sRNA; (**b**) KEGG enrichment of target genes of MvsH-expressed sRNA.

**Figure 11 microorganisms-11-02401-f011:**
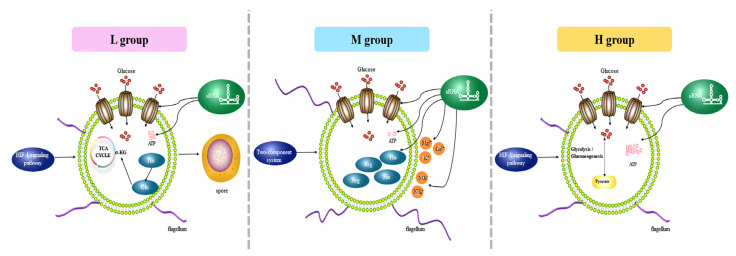
Adaptation mode of *B. subtilis* BS-G1 under different glucose concentrations.

**Figure 12 microorganisms-11-02401-f012:**
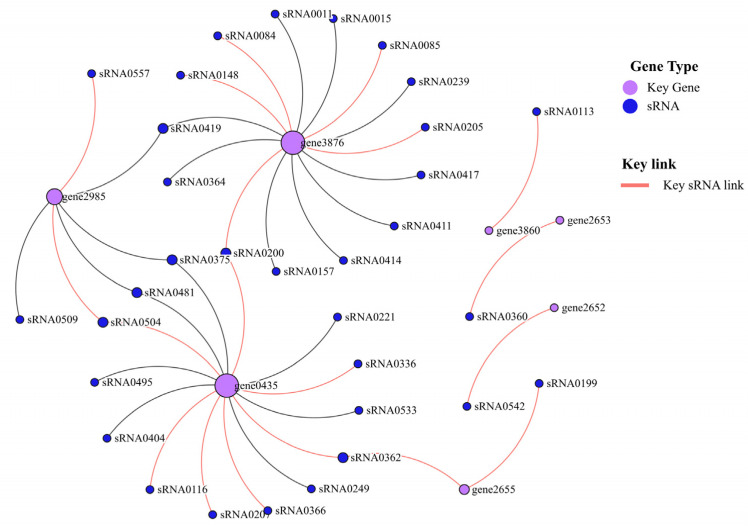
Crucial mRNAs and sRNAs interaction network diagram.

**Table 1 microorganisms-11-02401-t001:** Critical genes (mRNA) adapted to L group.

Key Gene ID	Description	Gene Name	Log_2_FC (LvsH)
gene2652	amino acid ABC transporter ATPase	/	7.448
gene2985	ABC transporter permease	/	10.238
gene3860	S	*celB*	9.383
gene0435	PTS mannitol transporter subunit IIB	*mtlA*	7.98
gene3876	oligo-beta-mannoside permease IIC protein	*celB*	7.362
gene2653	glutamine ABC transporter substrate-binding protein	/	7.496
gene2655	glutamine ABC transporter permease	/	5.051

**Table 2 microorganisms-11-02401-t002:** Critical mRNAs and sRNAs adapted to L group.

sRNA_id	Log_2_FC (LvsH)	Target Gene ID
sRNA0200	11.71890433	gene0435; gene3876
sRNA0113	11.533039	gene3860
sRNA0084	11.40169888	gene3876
sRNA0085	11.40169888	gene3876
sRNA0148	11.40169888	gene3876
sRNA0207	10.90597229	gene0435
sRNA0205	9.776851885	gene3876
sRNA0366	8.533373277	gene0435
sRNA0336	6.681651624	gene0435
sRNA0362	6.434679786	gene0435; gene2655
sRNA0116	6.373911204	gene0435
sRNA0360	6.357940855	gene2653
sRNA0542	6.212579528	gene2652
sRNA0557	6.026353075	gene2985
sRNA0199	5.159987109	gene2655
sRNA0504	5.048301125	gene2985; gene0435
sRNA0015	4.964673587	gene3876
sRNA0533	4.896095911	gene0435
sRNA0419	4.801798083	gene2985; gene3876
sRNA0239	4.618899947	gene3876
sRNA0249	4.578804747	gene0435
sRNA0495	3.975578933	gene0435
sRNA0411	3.854669068	gene3876
sRNA0509	3.764795398	gene2985
sRNA0221	3.520639327	gene0435
sRNA0414	3.022561069	gene3876
sRNA0481	2.967066947	gene0435; gene2985
sRNA0417	1.934699587	gene3876
sRNA0364	1.373518955	gene3876
sRNA0011	1.103524326	gene3876
sRNA0157	−1.52566018	gene3876
sRNA0404	−1.81557978	gene0435
sRNA0375	−2.395421117	gene2985; gene0435

## Data Availability

The RNA-Seq data are available at the NCBI Sequence Read Archive (SRA) under BioProject PRJNA 1011444 (SRA accession number SRR25890184, SRR25890183, SRR25890177, SRR25890170, SRR25890182, SRR25890181, SRR25890180, SRR25890179, SRR25890178. Sample accession numbers, SAMN37221208, SAMN37221209, SAMN37221210, SAMN37221217, SAMN37221218, SAMN37221219, SAMN37221220, SAMN37221221, SAMN37221222).

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
