# Peer review of "Understanding the Effect of Different Glucose Concentrations in the Oligotrophic Bacterium Bacillus subtilis BS-G1 through Transcriptomics Analysis"

_microorganisms, 2023, doi:10.3390/microorganisms11102401_

Round 1
Reviewer 1 Report
This is a manuscript that investigates the effect of substrate concentration on oligotrophic bacteria through transcriptome analysis. The experimental methods were very well executed. In particular, conducting the study with facultative oligotrophic bacteria led to obtaining intriguing results. There are a few issues however that I think need to be cleaned up in this manuscript.
Research on obligate and facultative oligotrophic bacteria has been to some extent explored. The bacterium mentioned by the author, Planctomycetes, is better suited for investigations into uncultured microbes rather than nutrient-poor conditions. PTS is known to be crucial for bacterial glucose transportation, so changes due to glucose concentration are expected outcomes and not particularly remarkable. There is no need to emphasize this point. Providing information about the changes in response to environmental conditions, especially concerning sRNAs, would be beneficial.
Overall, there are too many figures, and supplementary data are extensive, making it challenging to understand the results. Also, there are discrepancies in figure numbering between the main text and the figures themselves, which should be addressed. Transcriptome analysis heavily depends on the timing of sampling in the culture. The author collected samples at the 8-hour time point, which seems somewhat early when examining the growth curve. Is there a specific reason for this early sampling?
The results section is divided into many subsections, making it somewhat challenging to grasp the findings. Therefore, an overall summary and consolidation of the results are necessary.
The conclusion lacks sufficient discussion on the hypothesis the author wants to assert. It needs to be more concise, focusing on the core aspect of this study, which is the differences in mRNA and sRNA due to substrate concentration. Reducing unnecessary descriptions and providing a more in-depth analysis of these differences is essential.
Minor comments:
The methods section is quite poor compared to the experimental results.
Please summarize the content from “Line 101 to 126.”
Ensure consistency in using either "PTS" or "PTS system."
Why is the OD value decreasing for the five strains that used glucose? Is this due to a low quantity of glucose? (Figure1A, Figure 2A)
It seems that "LvsH" is an abbreviation for "Low vs High." It would be helpful to mention this abbreviation initially.
There is an overwhelming amount of data, making it difficult to focus. Additionally, there are several inaccuracies that need correction. (In Line 314, change "Figure 6B" to "Figure 7." Please check and correct other parts as well.)
Italicize bacterial strain names.
Author Response
Response to Reviewer 1 Comments
|
||
1. Summary |
|
|
Thank you very much for taking the time to review this manuscript. Please find the detailed responses below and the corresponding revisions/corrections highlighted/in track changes in the re-submitted files. |
||
2. Questions for General Evaluation |
Reviewer’s Evaluation |
Response and Revisions |
Does the introduction provide sufficient background and include all relevant references? |
Can be improved |
I give your corresponding response in the point-by-point response letter. The same as below |
Are all the cited references relevant to the research? |
Can be improved |
|
Is the research design appropriate? |
Yes |
|
Are the methods adequately described? |
Must be improved |
|
Are the results clearly presented? |
Must be improved |
|
Are the conclusions supported by the results? |
Must be improved |
|
3. Point-by-point response to Comments and Suggestions for Authors |
||
Comments 1: [There are too many figures, and supplementary data are extensive, making it challenging to understand the results.] |
||
Response 1: Thank you for pointing this out. We agree with this comment. Therefore, To minimize the number of figures within the primary body of the text, Figure 13 has been modified to Figure S3 in the article. In addition, we simplified 31 sub-tables to 16 sub-tables. I have implemented alterations to the content found on the first paragraph of page 14, within the range of lines 402. The modified section has been indicated in red. |
||
Comments 2: [there are discrepancies in figure numbering between the main text and the figures themselves, which should be addressed.] |
||
Response 2: Agree. We have, accordingly, modified the figure numbering between the main text. The modified section has been indicated in red. We changed "Figure 2A" to "Figure 2a". The revised part is in the third paragraph of page 5, line 211. We changed "Figure 2B" to "Figure 2b". The revised part is in the third paragraph of page 5, line 224. We added "Table S2" in the article. The revised part is in the first paragraph of page 6, line 242. We changed "Figure 3A, Figure 3B" to "Figure 3". The revised part is in the second paragraph of page 6, line 257. We changed "Table S8" to "Table S6". The revised part is in the second paragraph of page 7, line 278. We changed "Table S9" to "Table S7". The revised part is in the first paragraph of page 8, line 281. We changed "Table S10" to "Figure 6a". The revised part is in the second paragraph of page 8, line 285. We changed "Figure 6" to "Figure 6b". The revised part is in the second paragraph of page 8, line 287. We changed "Figure 6B" to Figure "7b". The revised part is in the first paragraph of page 9, line 300. We changed "Figure 7A" to Figure "7a". The revised part is in the first paragraph of page 9, line 305. We added "Figure 7c" in the article. The revised part is in the first paragraph of page 9, line 307. We changed "Figure 9A and 9B", to "Figure 9". The revised part is in the first paragraph of page 10, line 321. We changed "Figure 9A", to "Figure 9a". The revised part is in the first paragraph of page 10, line 324. We changed "Figure 9B", to "Figure 9b". The revised part is in the first paragraph of page 10, line 326. We changed "Table S12", to "Table S8". The revised part is in the second paragraph of page 10, line 337. We changed " Tables 1 and S13", to "Table S9". The revised part is in the second paragraph of page 10, line 341. We added "Figure 10a" in the article. The revised part is in the second paragraph of page 10, line 343. We changed " Tables S14-S15 and Figure 10", to "Figure 10b". The revised part is in the second paragraph of page 10, line 346. We changed " Figure 13", to "Figure S3". The revised part is in the first paragraph of page 14, line 402. We added "Figures 5 and 6" in the article. The revised part is in the second paragraph of page 14, line 411. We changed " Tables S28 and S29", to " Tables S10 and S11". The revised part is in the third paragraph of page 14, line 435. We added "Tables S10 and S11" in the article. The revised part is in the first paragraph of page 15, line 455. We changed " Tables S28 and S29", to " Tables S10 and S11". The revised part is in the second paragraph of page 15, line 464. We changed " Tables S32 and S33", to " Tables S15 and S16". The revised part is in the first paragraph of page 16, line 507. Comments 3: [Transcriptome analysis heavily depends on the timing of sampling in the culture. The author collected samples at the 8-hour time point, which seems somewhat early when examining the growth curve. Is there a specific reason for this early sampling?] Response 3: Thank you for pointing this out. We agree with this comment. In prior experiments, the cultivation of strains for durations of 9-16 hours yielded a culture medium exhibiting excessive viscosity, thereby leading to pronounced RNA contamination and subsequent failure in transcriptome sequencing. After multiple tests and attempts, we have chosen the optimal cultivation time (8 hours). When the strain is cultured for 8 hours, high-quality total RNA can be obtained. Then proceed with transcriptome sequencing. Comments 4: [The results section is divided into many subsections, making it somewhat challenging to grasp the findings. Therefore, an overall summary and consolidation of the results are necessary.] Response 4: Agree. We have, accordingly, merged the experimental results. ("3.3. Functional annotation and expression analysis of transcripts" and " 3.3. Identification of DEGs" were merged into "3.2. Transcriptome analysis of B. subtilis under different glucose concentrations"). The revised part is in the first paragraph of page 6, lines 229 to 250. The modified section has been indicated in red. Comments 5: [The conclusion lacks a sufficient discussion on the hypothesis the author wants to assert. It needs to be more concise, focusing on the core aspect of this study, which is the differences in mRNA and sRNA due to substrate concentration. Reducing unnecessary descriptions and providing a more in-depth analysis of these differences is essential.] Response 5: Thank you for pointing this out. We agree with this comment. Modifications have been implemented to the conclusion section, with a focus on investigating the impact of varying glucose concentrations on the expression of strain mRNA and sRNAs. The revised part is in the second and third paragraph of page 17, lines 565 to 593. The modified section has been indicated in red. Comments 6: [The methods section is quite poor compared to the experimental results.] Response 6: Agree. We have, accordingly, modified the experimental method. Three contents have been added to the experimental method. (" 2.4. Reading Mapping and Analysis", "2.6. Quantitative Real-Time PCR (qRT-PCR)" and"2.7. Statistical Analysis") The revised part is in the five paragraph of page 4, lines 180-190. The revised part is in the sixth and seventh paragraph of page 4, lines 195-201. The revised part is in the eighth paragraph of page 4, lines 202-206. The modified section has been indicated in red. Comments 7: [Please summarize the content from “Line 101 to 126.”] Response 7: Thank you for pointing this out. We agree with this comment. This sentence comes from the doctoral thesis of Ligui Wang (Chinese Academy of Military Medical Sciences, PLA) This sentence is "Studies have shown that approximately 10-30% of bacterial genes are regulated by " Due to the lack of support from other literature, I have deleted this sentence. ( Title of Ligui Wang's paper: Discovery of a novel sRNA of Shigella flexneri based on RNA Seq technology and exploration of its pathogenic mechanism) The sentence is" Many researchers have comprehensively studied the response of oligotrophic bacteria to low nutrition." This sentence is not precise, so I have made modifications to it and cited two references. We will change this sentence to" Many researchers have studied the response of oligotrophic bacteria to low nutrition. '' The revised part is in the second paragraph of page 3, lines 101-110. The modified section has been indicated in red. Comments 8: [Ensure consistency in using either "PTS" or "PTS system.] Response 8: Agree. We have, accordingly, changed the PTS system to PTS in the article. There are a total of 30 PTS in the article. The modified section has been indicated in red. Comments 9: [Why is the OD value decreasing for the five strains that used glucose? Is this due to a low quantity of glucose? (Figure1A, Figure 2A)] Response 9: Thank you for pointing this out. We agree with this comment. In this culture medium, glucose is the only carbon source. Because a glucose concentration of 0.015g/L is an extremely low glucose concentration. Extremely low glucose concentration will inhibit the growth of the strain, so the OD value decreases for the five strains. Comments 10: [It seems that "LvsH" is an abbreviation for "Low vs. High." It would be helpful to mention this abbreviation initially.] Response 10: Thank you for pointing this out. We agree with this comment. "LvsH" is an abbreviation for "L group vs. H group". "LvsH": Using the H group as the control, compare the L group with it to obtain the differential expression between the H group and the L group. I have annotated the meaning of "LvsH" in the article. The revised part is in the first paragraph of page 6, line 245. The modified section has been indicated in red. Comments 11: [There is an overwhelming amount of data, making it difficult to focus. Additionally, several inaccuracies need correction. (In Line 314, change "Figure 6B" to "Figure 7." Please check and correct other parts as well.).] Response 11: Agree. We have, accordingly, changed "Figure 6B" to "Figure 7b." in the article. The revised part is in the first paragraph of page 9, line 300. We have checked the article and made revisions. The modified section has been indicated in red. Comments 12: [Italicize bacterial strain names.] Response 12: Thank you for pointing this out. We agree with this comment. I have set all strains in the text to italics. The modified section has been indicated in blue. |
||
4. Response to Comments on the Quality of English Language |
||
Point 1: I am not qualified to assess the quality of English in this paper. |
||
Response 1: I have uploaded this article to the website for polishing. I have made revisions to the article based on the polished results. (https://www.mdpi.com/authors/english) |
||
5. Additional clarifications Response: Thank you for your suggestion. I don't have any other explanation. |

Reviewer 2 Report
The manuscript titled 'Understanding the effect of different glucose concentrations in the oligotrophic bacterium Bacillus subtilis BS-G1 through transcriptomics analysis' is devoted to the exploration of the mechanisms underlying the adaptation of oligotrophic bacteria to nutrient-poor environments. Following revision, the manuscript might be of interest for the readership of Microorganisms.
Reviewer Comments:
1. The manuscript exhibits a number of typographical errors, formatting inconsistencies, and other mistakes. A thorough proofreading process is strongly recommended to rectify these issues.
2. The introduction section requires refinement to provide a clearer contextual description. Emphasizing the novelty and providing a comprehensive background for the study is recommended. Furthermore, it is suggested to separate and delve into detailed analyses of studies involving Bacillus and other oligotrophic bacteria.
3. All figures should be uniformly formatted and the font size should be consistent throughout. Several figures are currently unreadable and should be addressed accordingly.
4. The construction principle behind the LvsH and MvsH groups needs clarification for improved understanding.
5. The manuscript would benefit from experimental verification of the involvement of key genes in glucose metabolism.
The manuscript exhibits a number of typographical errors, formatting inconsistencies, and other mistakes.
Author Response
Response to Reviewer 2 Comments
|
||
1. Summary |
|
|
Thank you very much for taking the time to review this manuscript. Please find the detailed responses below and the corresponding revisions/corrections highlighted/in track changes in the re-submitted files. |
||
2. Questions for General Evaluation |
Reviewer’s Evaluation |
Response and Revisions |
Is the work a significant contribution to the field? |
[I want to give my corresponding response in the point-by-point response letter. The same as below] |
|
Is the work well organized and comprehensively described? |
|
|
Is the work scientifically sound and not misleading? |
|
|
Are there appropriate and adequate references to related and previous work? |
|
|
Is the English used correct and readable? |
|
|
3. Point-by-point response to Comments and Suggestions for Authors |
|
|
Comments 1: [The manuscript exhibits a number of typographical errors, formatting inconsistencies, and other mistakes. A thorough proofreading process is strongly recommended to rectify these issues.] |
||
Response 1: Thank you for pointing this out. We agree with this comment. Therefore, we have made revisions to the manuscript due to many printing errors, inconsistent formatting, and other errors. (1) We have changed all DEG to DEGs in the article. (2) I will modify the error in the title of Figure 2. Original text: Figure 2 Glucose uptake curve (A) Glucose uptake curves of B. subtilis BS-G1 and 5 control strands at 0.015g/L-10g/L glucose concentration uptake curve; After modification: Figure 2 Glucose uptake curve (a) Glucose uptake curves of B. subtilis BS-G1 and 5 control chains at 0.015g/L glucose concentration uptake curve. The revised part is in the first paragraph of page 6, lines 226-227. (The modified section has been indicated in red.) |
||
Comments 2: [The introduction section requires refinement to provide a clearer contextual description. Emphasizing the novelty and providing a comprehensive background for the study is recommended. Furthermore, it is suggested to separate and delve into detailed analyses of studies involving Bacillus and other oligotrophic bacteria.] |
||
Response 2: Thank you for pointing this out. We agree with this comment. We have added research on Bacillus subtilis in oligotrophic aspects. The revised part is in the second paragraph of page 3, lines 110-125. In addition, we have added relevant research on oligotrophic bacteria. The revised part is in the second paragraph of page 3, lines 101-110. The modified section has been indicated in red. Comments 3: [All figures should be uniformly formatted and the font size should be consistent throughout. Several figures are currently unreadable and should be addressed accordingly.] Response 3: Agree. We have, accordingly, changed unified the charts and fonts in the article. In addition, I have corrected the numbering in the chart. According to the format requirements of the image, I have changed the uppercase letters in the image and image titles to lowercase letters. For example, changing (A) to (a). Modifications to Figure 1 and Figure Title: The revised part is in the second paragraph of page 5, lines 215-218. Modifications to Figure 2 and Figure Title: The revised part is in the first paragraph of page 6, lines 225-227. Modifications to Figure 3 、Figure 4 and Figure Title: The revised part is in the first paragraph of page 7, lines 264-270. Modifications to Figure 5 and Figure Title: The revised part is in the second paragraph of page 8, lines 288-290. Modifications to Figure 6 and Figure Title: The revised part is in the second paragraph of page 8, lines 294-297. Modifications to Figure 7 and Figure Title: The revised part is in the first paragraph of page 9, lines 311-314. Modifications to Figure 9 and Figure Title: The revised part is in the first paragraph of page 10, lines 326-330. Modifications to Figure 10 and Figure Title: The revised part is in the first paragraph of page 11, lines 349-352. The modified section has been indicated in red. Comments 4: [The construction principle behind the LvsH and MvsH groups needs clarification for improved understanding.] Response 4: Thank you for pointing this out. We agree with this comment. "LvsH": Using the H group as the control, compare the L group with it to obtain the differential expression between the H group and the L group. "MvsH": Using the H group as the control, compare the M group with it to obtain the differential expression between the H group and the M group. "LvsH" in the article. The revised part is in the first paragraph of page 6, lines 246-247. The modified section has been indicated in red. Comments 5: [The manuscript would benefit from experimental verification of the involvement of key genes in glucose metabolism.] Response 5: Thank you for your suggestion. We agree with this comment. It is very important. Due to your suggestion, I have discovered the shortcomings of my current work. I will improve my research level and achieve more results in my future work according to your suggestion. |
||
4. Response to Comments on the Quality of English Language |
||
Point 1: The manuscript exhibits a number of typographical errors, formatting inconsistencies, and other mistakes. |
||
Response 1: I have uploaded this article to the website for polishing. I have made revisions to the article based on the polished results. (https://www.mdpi.com/authors/english) I have corrected the formatting errors in the manuscript. |
||
5. Additional clarifications |
||
Thank you for your suggestion. We have used homologous recombination to knock out key genes, but the knockout strains cannot survive. |

Round 2
Reviewer 1 Report
I enjoyed reading the revised version of this manuscript and I would like to thank the authors for such a careful revision. I am convinced that this manuscript contains important information for all researchers interested in the same field.
Reviewer 2 Report
The revised version of the manuscript is significantly improved and I find it suitable for publication in Microorganisms.